# Cytoreductive Surgery with Hyperthermic Intraperitoneal Chemotherapy for Peritoneal Carcinomatosis from Epithelial Ovarian Cancer: A 20-Year Single-Center Experience

**DOI:** 10.3390/cancers13030523

**Published:** 2021-01-29

**Authors:** Fabio Carboni, Orietta Federici, Isabella Sperduti, Settimio Zazza, Domenico Sergi, Francesco Corona, Mario Valle

**Affiliations:** 1Peritoneal Tumours Unit, IRCCS Regina Elena National Cancer Institute, 00144 Rome, Italy; orietta.federici@ifo.gov.it (O.F.); settimio.zazza@ifo.gov.it (S.Z.); francesco.corona@ifo.gov.it (F.C.); mario.valle@ifo.gov.it (M.V.); 2Biostatistical Unit, IRCCS Regina Elena National Cancer Institute, 00144 Rome, Italy; isabella.sperduti@ifo.gov.it; 3Department of Oncology, IRCCS Regina Elena National Cancer Institute, 00144 Rome, Italy; domenico.sergi@ifo.gov.it

**Keywords:** ovarian cancer, peritoneal carcinomatosis, cytoreductive surgery, HIPEC, results

## Abstract

**Simple Summary:**

Multimodality treatment is the standard treatment for epithelial ovarian cancer, but the peritoneum is the primary site of spread or relapse in most cases. Cytoreductive surgery with hyperthermic intraperitoneal chemotherapy has been introduced in order to improve outcomes, but most studies, including both primary and recurrent cases, are retrospective, non-randomized and heterogeneous. The aim of this study was to report a 20-year single-center experience with this treatment. In our study, it appeared to be a feasible procedure, with acceptable postoperative morbidity and mortality rates, providing different survival benefits depending on the timing of surgery, as long as a complete cytoreduction was obtained. Until the results from ongoing prospective randomized clinical trials clarify the role and appropriate indications, cytoreductive surgery with hyperthermic intraperitoneal chemotherapy may be considered an effective treatment for selected cases of epithelial ovarian cancer, if performed in specialized centers.

**Abstract:**

Despite improvement in treatments, the peritoneum remains the primary site of relapse in most ovarian cancer cases. Patients who underwent cytoreductive surgery with hyperthermic intraperitoneal chemotherapy for peritoneal carcinomatosis from epithelial ovarian cancer were reviewed. Kaplan–Meier curves and multivariate Cox analyses were used to identify survival rates and prognostic factors. This study included 158 patients. The procedure was mostly performed for recurrent disease (46.8%) and high-grade serous carcinoma (58.2%). The median peritoneal cancer index was 14, and complete cytoreduction was obtained in 87.9% of cases. Grade IV morbidity occurred in 15.2% of patients, mostly requiring surgical reoperation, and one patient (0.6%) died within 90 days. The median follow-up was 63.5 months. The Kaplan–Meier 5-year overall survival (OS) and disease-free survival (DFS) rates were 42.1% and 24.3%, respectively. Multiple regression logistic analyses demonstrated that the completeness of cytoreduction (CC) score (*p* ≤ 0.0001), pancreatic resection (*p* ≤ 0.0001) and number of resections (*p* = 0.001) were significant factors influencing OS; whereas the CC score (*p* ≤ 0.0001) and diaphragmatic procedures (*p* = 0.01) were significant for DFS. The addition of hyperthermic intraperitoneal chemotherapy to standard multimodality therapy may improve outcomes in both primary and recurrent epithelial ovarian cancer without impairing early postoperative results, but the exact timing has not yet been established. Prospective randomized studies will clarify the role and indications of this approach.

## 1. Introduction

Regardless of its infrequent incidence, epithelial ovarian cancer (EOC) is the fourth leading cause of cancer death in women, and the most lethal gynecological malignancy, accounting for an estimated 295,414 new cases and 184,799 deaths worldwide in 2018 [1]. Due to the lack of specific symptoms, approximately two-thirds of patients show an advanced stage at diagnosis, and the 5-year survival rate is less than 30% [2,3]. The term ovarian cancer includes multiple distinct entities, but approximately 90% arise from the surface layer of the ovaries or the lining of the fallopian tubes, and pathological assessment is essential in selecting the most appropriate treatment.

The mainstay of treatment is aggressive cytoreductive surgery (CRS) in order to remove all macroscopic disease, because the absence of residual disease is the most important prognostic factor. Despite an improvement in treatment strategies, the peritoneum remains the primary site of spread and relapse in most cases [4,5]. There is increasing evidence that the addition of hyperthermic intraperitoneal chemotherapy (HIPEC) to CRS improves patient prognosis, however, its role is still under debate [6,7,8,9,10]. We report our 20-year single-center experience on CRS with HIPEC for peritoneal carcinomatosis (PC) from EOC.

## 2. Results

A total of 158 patients were identified in the period of this study. Baseline characteristics of patients and perioperative findings are described in Table 1.

The median age was 58 years, and high-grade serous tumors comprised the majority of cases (58.2%). About half of the patients had undergone previous surgery, and 40% of them underwent treatment for primary disease in upfront or interval debulking surgery (IDS). The median peritoneal cancer index (PCI) was 14 and CC0–1 resection was obtained in 87.9% of cases. Extended surgery, including major abdominal procedures, was performed in the majority of patients, with a mean of 12 resected regions for each case. More than 11 procedures were performed in 63.9% of cases (101 patients). Lymph node dissection was performed in 54.4% of patients, 40.7% of which were positive. The mean length of stay was 25 days (range 0–77). The types of surgical procedures are depicted in Table 2.

Based on the Common Terminology Criteria for Adverse Events (CTCAE), 85 (53.8%) patients presented no adverse events, whereas 49 (31%) had grade I–II complications. No grade III adverse events occurred, but 24 patients (15.2%) qualified as having grade IV complications (Table 2). A surgical reoperation was required in 23 cases, whereas one case of severe pneumonia required intensive care unit (ICU) readmission. In-hospital mortality (grade V) rate was 0.6%: a 60-year-old woman with severe obesity (BMI 39) who underwent reoperation for abdominal bleeding died of multi-organ failure 60 days after surgery.

The median follow-up was 63.5 months (CI95% 43.5–83.5). The median overall (OS) and disease-free (DFS) survival for the entire group was 52 (CI95% 43–62) and 20 (15–25) months, respectively. Among the perioperative factors identified at Cox analysis and at univariate analysis for OS, only the completeness of cytoreduction (CC) score (*p* < 0.0001), pancreatic resection (*p* = 0.06) and number of resections (*p* = 0.08) remained significant at multivariate analysis (Table 3).

Among those associated with DFS at univariate analysis, only the CC score (*p* < 0.0001) and diaphragmatic procedures (*p* = 0.01) were significant at multivariate analysis (Table 4).

Kaplan–Meier 5-year OS and DFS rates were 42.1% and 24.3%, respectively (Table 5). 

The CC score (*p* ≤ 0.0001) (Figure 1A), pancreatic resection (*p* ≤ 0.0001) (Figure 1B) and number of resections (*p* = 0.001) (Figure 1C) were significant factors influencing OS.

The CC score (*p* ≤ 0.0001) (Figure 2A) and diaphragmatic procedures (*p* = 0.01) (Figure 2B) were significant factors influencing DFS.

A subgroup analysis was conducted on 86 patients who underwent lymph node dissection, combining their CC score with their lymph node status. A similar DFS was observed between CC1 and CC2 resections (13 vs. 11 months) in patients with negative lymph nodes (Figure 3A), whereas it was improved without significance in patients with positive lymph nodes (39 vs. 10 months) (Figure 3B).

The OS was improved without significance in CC1 with respect to CC2 resections (39 vs. 11 months) in patients with positive lymph nodes (Figure 4A), as well as for CC0 with respect to CC1 resections (40 vs. 72 months) in the group of patients with negative lymph nodes (Figure 4B).

Five-year survival results were finally evaluated according to the different timing of procedure. In the upfront and IDS subgroups of patients, DFS was improved without significance with respect to recurrent and salvage subgroups, whereas the difference in OS resulted as significant (*p* = 0.03) (Figure 5).

## 3. Discussion

EOC represented the most common indication among a total of approximately 400 CRSs with HIPEC performed over a 20-year period. CC0 resection resulted the most significant prognostic factor, influencing both OS and DFS, and extended surgery was required in the majority of our patients. Due to the routine use of preoperative laparoscopic staging and the introduction of a standardized protocol for the prevention, monitoring and treatment of infections [11], a selection of patients and postoperative outcomes have been improved during the years. With the growth of our experience, we observed that the small bowel and its mesentery affected resectability more than total PCI score. The timing of the procedure influenced survival rates, with the IDS group showing the best results, confirming that a multidisciplinary approach is the key to obtain the improvement of results.

Despite excellent response rates to front-line multimodality therapy, including primary debulking surgery followed by intravenous chemotherapy (CHT) or IDS [12,13,14,15], recurrence confined to the peritoneal cavity occurs in up to 70% of patients within 5 years [2,3,4,16]. Since the peritoneum is the primary site of spread and failure in relapses, intraperitoneal chemotherapy has been administered with efficacy in optimally debulked advanced cases, but, criticisms due to the clinical trial’s design, higher toxicity, catheter complications and discontinuity prevented this treatment from being widely adopted in clinical practice [1,3,17]. The addition of HIPEC at the end of CRS may offer several potential advantages for treating residual microscopic disease that cannot be significantly managed. The role in EOC is still unclear, since most studies, including both primary and recurrent cases, are retrospective, non-randomized and heterogeneous [6,7,8,9,10]. Postoperative morbidity and mortality rates have decreased considerably in high-volume centers, and our 0.6% and 15.2% incidence rates of postoperative mortality and grade IV morbidity are in line with reported results. The majority of complications are due to the aggressive surgery, whereas the addition of HIPEC may be associated with hematologic toxicity due to transient renal impairment and bone marrow immunosuppression [7,8,9]. Selected extended procedures influenced OS in our series, but only pancreatic resection and the number of resections remained significant prognostic factors, together with the CC score at multivariate analysis. The same occurred for DFS, but only diaphragmatic procedures and the CC score were significant at multivariate analysis. Diaphragmatic surgery is often required in patients with EOC in order to obtain complete CRS. In our recently reported experience, it was associated with a significant incidence of postoperative complications, especially bleeding [18]. Involvement of the diaphragm usually reflects advanced disease and is correlated to a higher PCI score, probably explaining the decreased DFS observed in this series. PCI was a significant factor for both OS and DFS only at univariate analysis in our series. In agreement with other authors, we believe that the extent of the involvement of the small bowel and its mesentery is more relevant than the total PCI score, since it represents the true key point in determining complete resectability [19,20].

The first large prospective randomized phase III trial on IDS with HIPEC for the treatment of primary EOC reported increased OS and PFS, without higher incidence of side effects [21]. The improvement of OS and DFS rates in the treatment of primary disease has been confirmed in several systematic reviews and meta-analyses [7,8,9,22]. Decreased OS and DFS rates have been observed with IDS after no response to neoadjuvant chemotherapy (NACT) compared to patients who underwent upfront surgery [23,24]. Timing proved significant at univariate analysis in our series. In our opinion, when a patient is candidate to IDS with HIPEC, preoperative laparoscopic staging in a patient candidate to IDS with HIPEC should only include biopsies, avoiding adnexectomy, since wide ligaments of the uterus opening may give rise to extraperitoneal spreading of the disease. The use of HIPEC in addition to upfront surgery is particularly attractive but delicate since the effectiveness of CHT has been widely demonstrated. In our experience, the upfront and IDS subgroup of patients showed significantly higher OS rates (56% and 57%, respectively), but the low number of cases limits the statistical power. The OVHIPEC-2 trial has been designed and will hopefully answer whether HIPEC during upfront CRS is beneficial [25].

The benefit of lymph node dissection is unclear, but it does not seem to play a role in the improvement of outcomes [15,26] and it was performed for staging purposes in our series. Incomplete resection and the presence of positive lymph nodes have been identified as risk factors of early recurrence within 1 year in a large single-center retrospective study [27]. According to the authors, when suboptimal CRS is anticipated and bulky lymph nodes are present at preoperative work-up, patient selection should be very cautious. The results obtained in our subgroup of patients who underwent lymph node dissection are worthy of reflection. In the case of negative lymph nodes, the CC score was the strongest predictor of OS and DFS, but similar DFS rates between CC1 and CC2 resections were observed. A possible explanation is that adjuvant CHT was always administered after CC2 resection, whereas it was administered only in the case of recurrence after CC1 resection. In patients with positive lymph nodes, OS and DFS rates did not differ significantly between CC0 and CC1 resections, probably because adjuvant CHT was always administered in both cases.

Approximately half of our cases were constituted by recurrent EOC. Cytoreduction has been widely adopted in the treatment of the disease and there has always been a big interest in HIPEC, but its role has still not been defined. The greatest benefit has been demonstrated in patients with no residual disease and those considered platinum-sensitive [28,29,30]. No consensus exists for a specific chemotherapy agent or protocol to be used, and evidence is currently limited to single institution experiences and retrospective studies. Systematic reviews confirmed that only OS is improved in patients with recurrent disease [7,8,9], but subgroup analysis revealed that HIPEC also improved DFS in patients who received optimal cytoreduction [22]. Five-year DFS and OS rates were 18% and 32%, respectively, in this series. HIPEC may overcome systemic resistance to platinum agents in this setting, but results are controversial and the question is open [22,25,29,30]. Prolonged NACT is believed to be a risk factor for platinum-resistance in selected histologic subtypes, which may give rise to a less sensitive recurrent disease [31,32]. In our experience, when a disease-free interval more than 6 months after CC0 resection is achieved, the same first line regimen of CHT may be administered with efficacy if required. The results of the CHIPOR trial will provide new insights about the effects of HIPEC in the recurrent setting [25].

Although CC1 resection is considered optimal, the patient’s prognosis is significantly lower than CC0 resection. The CC0 score was the strongest predictor of survival for both OS and DFS in our series (*p* ≤ 0.0001). The median OS and DFS rates were 74 and 38 months after CC0 resection and 39 and 13 months after CC1, respectively. The definition of residual disease after surgery has evolved over time, but remains controversial and often differs among studies. Optimal cytoreduction according to the CC score corresponds to tumor residual less than 0.25 cm (CC1), whereas previously adopted criteria included residual disease up to 1 cm, corresponding to a CC2 score [33,34,35]. Intraoperative estimation of PCI is currently considered the gold standard for evaluating the extent of disease, but cytoreduction may be underrated after NACT, since it alters the morphology of peritoneal deposits, making it difficult to evaluate the extent of dissemination. A high incidence of occult microscopic disease in normal-looking areas of peritoneum and scar tissues occur, which may be left inside in up to half of the cases [34,36]. The complete removal of parietal peritoneum and “target regions” should always be performed, regardless of the presence of visible disease. An increased incidence of scar tissues with an increasing number of NACT cycles was observed in our experience, especially in the diaphragm. The impact of the number of NACT cycles administered before IDS on patient survival, and consequently the optimal timing for IDS, is still being debated. Poorer prognosis for patients receiving late IDS (after >4 CHT cycles) compared to those operated on earlier has been reported [13]. In accordance with Tate et al., we believe that the increased number of NACT cycles does not induce drug resistance, but it causes more lesions to become macroscopically invisible, allowing unresected pathological disease to remain on site, which may lead to drug resistance [37]. Our current protocol includes only three NACT cycles before the restaging of patients in the IDS group.

No standard treatment for patients with platinum-resistant and refractory disease has been developed in order to achieve long-term remission and maintain an acceptable quality of life. Most CHT regimes are very expensive, with high toxicity and minimal survival compared to the best supportive care [38]. The addition of HIPEC to CRS in this setting has rarely been described. Five-year OS and DFS rates of 32% and 9%, respectively, were obtained in our salvage group, without significant differences with respect to recurrent disease. In any case, HIPEC should not be performed if absence of macroscopic residual disease cannot be achieved.

Concerns about HIPEC include the potential increased cost associated, considering the length of stay and intensive care unit admissions, and the postoperative quality of life of the patients. Recent analyses showed that the procedure is cost-effective in the management of stage III OC [39,40,41] and health-related quality of life is not negatively affected, since side effects are typically resolved within a few months after treatment [42,43]. Because of the different regimens (drug, timing, duration, temperature) and the heterogeneity of populations (residual tumor, histology, stage) among investigations, there is controversy surrounding which patients can benefit from HIPEC. In agreement with most authors, we believe that HIPEC should not be considered an independent treatment modality, but as a complementary modality to systemic chemotherapy and novel targeted therapies [6,7,8,9,10,29,30,31,35,36,37,44].

The main limitations of this study include the heterogeneous cohort of patients, the inherent selection biases associated with the retrospective nature of the data and the relatively small number of cases. Despite this, it is an accurate description of a 20-year single-center experience.

## 4. Materials and Methods

A retrospective analysis of a prospectively maintained database was conducted on all patients who underwent CRS with HIPEC for PC from primary or recurrent epithelial ovarian, tubal or primary peritoneal cancer, between January 2000 and January 2020. In addition to standard cross-sectional imaging, preoperative laparoscopy was always performed in order to improve preoperative staging and patient selection [45]. The stage of the disease followed the current International Federation of Gynecology and Oncology (FIGO) classification system [46]. Surgical and HIPEC procedures were based on the technique originally described by Sugarbaker [47]. Principles of peritoneal surgery included: resection of the involved regions; resection of the so-called “target regions” even in the absence of visible disease (lesser omentum, gallbladder, falciform and umbilical round ligaments); omental resection, including gastroepiploic arch in the presence of visible disease. Bilateral iliac and obturator lymph node dissection was performed only in the case of bulky disease at the beginning of our experience, but it became routine thereafter in order to accurately stage the disease for possible adjuvant CHT. The extent of peritoneal involvement was classified according to the Peritoneal Cancer Index (PCI) and the completeness of cytoreduction (CC) score was used to assess residual tumor after surgery [47].

HIPEC was performed immediately after optimal cytoreduction (CC0–1). Body surface area (BSA)-based chemotherapy was administered according to a modified Sugarbaker protocol [48]. Cisplatin (100 mg/m^2^) and doxorubicin (30 mg/m^2^) were added to a 4000 mL isotonic saline solution. The perfusate was heated to a temperature of 42–42.5 °C and circulated into the peritoneal cavity for 90 min. The mean flow rate was 2000 mL/min and the global amount of perfusion was 4000 mL. Indications for CRS with HIPEC were categorized as follows: upfront, if performed as the first surgical therapy for primary disease; interval debulking surgery (IDS), if performed after neoadjuvant chemotherapy (NACT) for primary disease; recurrent, if performed for recurrent disease after upfront surgery or IDS; salvage, if performed after multiple surgical procedures or two or more unsuccessful lines of CHT for recurrent disease. In the interval group, patients received at least 3 or 4 cycles of NACT with a carboplatin–paclitaxel combination before restaging. If optimal cytoreduction was anticipated at laparoscopy, patients underwent CRS with HIPEC, otherwise 3 or 4 more cycles were administered. No antiangiogenic treatment (i.e., bevacizumab) in association with first-line CHT was administered. Systemic chemotherapy was always given after any type of surgery in the case of positive lymph nodes or CC2 resection, but only if recurrences occurred after CC0–CC1 resection. Two patients were considered platinum-resistant (i.e., recurrence occurring between 1 and 6 months after treatment) and both received doxorubicin-based CHT.

Postoperative results were categorized according to the 90-day morbidity and mortality National Cancer Institute’s Common Terminology Criteria for Adverse Events (NCI-CTCAE) version 4.0 [49]. A five-point scale to grade the severity of post-procedural adverse events was used. Clinical observation and minimal medical intervention were the only required treatments in minor and moderate complications (grade I and II). Grade III adverse events are defined as severe complications that are not immediately life-threating, but warrant imaging-guided percutaneous or surgical drainage (grade III). Life-threatening complications requiring urgent intervention are graded IV. Grade V is death related to adverse events.

Follow-up information was regularly obtained in all cases. For all outcomes, data collection started from the date of surgery and were censored at the time of the most recent follow-up or death. 

### Statistical Analysis

Demographic, clinical and operative variables were recorded for all patients. Descriptive statistics were adopted to summarize pertinent study information. The follow-up was analyzed with the reverse Kaplan–Meier method. Associations between categorial variables were analyzed with the Chi-square test or Fisher Exact test, when appropriate. Survival curves were estimated by the Kaplan–Meier product limit method. The log-rank test was used to assess differences between subgroups. Significance was defined at *p* < 0.05. The hazard ratio (HR) and the 95% confidence interval (95% CI) were estimated using the Cox univariate model. A multivariate Cox proportional hazard model was developed using stepwise regression (forward selection, enter limit and remove limit, *p* = 0.10 and *p* = 0.15, respectively), to identify independent predictors of outcomes. The Harrell’s guidelines for identifying the correct number of covariates were taken into account for the power analysis [50]. The SPSS (version 21.0) licensed statistical program was used for all analyses.

## 5. Conclusions

While new generation drugs and biological agents are increasingly introduced in clinical practice with promising results, the addition of HIPEC to standard multimodality therapy may improve outcomes in both primary and recurrent EOC. Although criticisms are present in many of the published trials, a clear trend toward survival benefit with an acceptable side effect profile is evident, but controversy remains surrounding which patients could benefit most from HIPEC. The results from the ongoing prospective randomized clinical trials will clarify the role and appropriate indications of this aggressive locoregional approach.

## Figures and Tables

**Figure 1 cancers-13-00523-f001:**
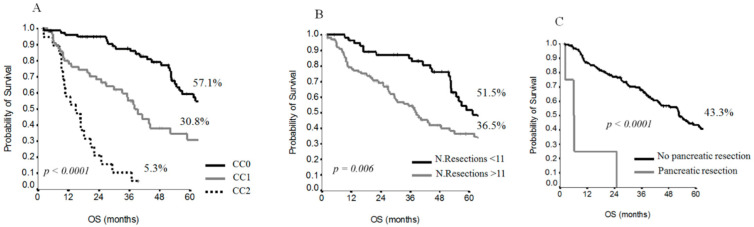
Survival curves for OS by CC score (**A**), number of resections (**B**) and pancreatic resection (**C**).

**Figure 2 cancers-13-00523-f002:**
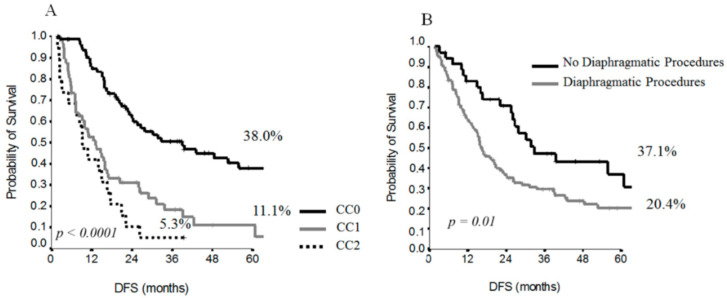
Survival curves for DFS by CC score (**A**) and diaphragmatic resection (**B**).

**Figure 3 cancers-13-00523-f003:**
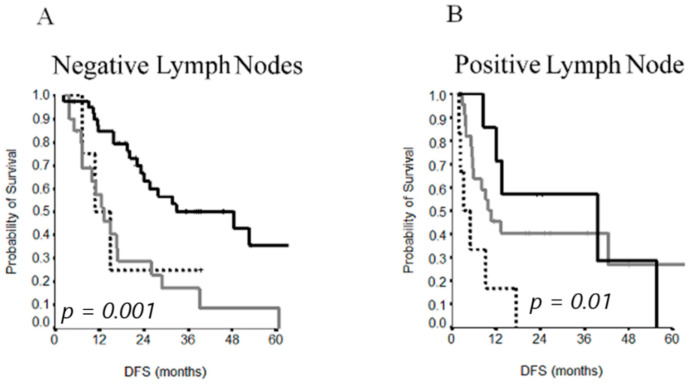
DFS curves by CC score (**A**) and lymph node status (**B**).

**Figure 4 cancers-13-00523-f004:**
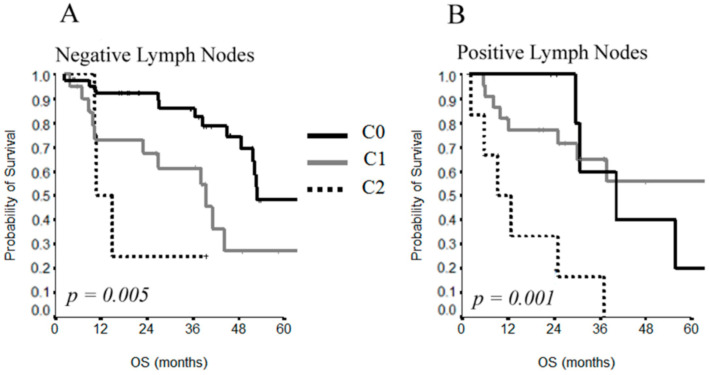
OS curves by CC score (**A**) and lymph node status (**B**).

**Figure 5 cancers-13-00523-f005:**
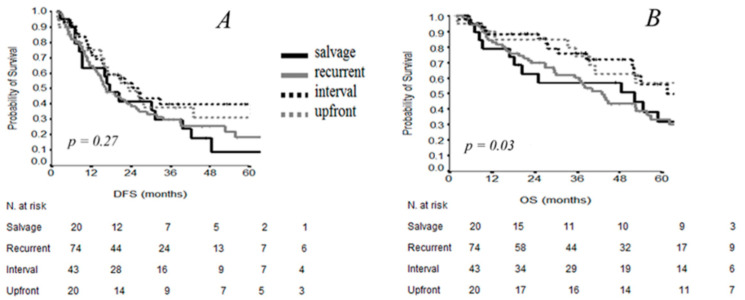
DFS (**A**) and OS (**B**) curves by timing of surgery.

**Table 1 cancers-13-00523-t001:** Baseline and perioperative features (total 158 patients).

Features	Results
Median Age, Years (Range)	58 (26–78)
Median BMI, kg/m^2^, n (range)	26 (17–39)
ASA II/III Class, n (%)	130 (82.3)
Previous surgery, n (%)	74 (46.8)
FIGO stage, n (%)	
II	1 (0.6)
IIIb	22 (13.9)
IIIc	135 (85.4)
Histology, n (%)	
High-grade serous	92 (58.2)
Primary peritoneal	22 (13.9)
Undifferentiated	36 (22.8)
Other	8 (5.1)
Type of surgery, n (%)	
Upfront	20 (12.7)
Interval	44 (27.8)
Recurrence	74 (46.8)
Salvage	20 (12.7)
PCI, median (range)	14 (3–34)
<15	87 (55.1%)
>15	71 (44.9%)
CC resection, n (%)	
CC0	83 (52.5)
CC1	56 (35.4)
CC2	19 (12)
Lymph node dissection, n (%)	86 (54.4)
Lymph nodes positive, n (%)	35 (40.7)
Mean Operative Time, min (range)	420 (240–600)
Resected regions, median (range)	12 (0–19)
Mean ICU stay, hours (range)	60 (12–120)
Median length of stay, days (range)	25 (0–77)

BMI, body mass index; ASA, American Society of Anesthesiologists; FIGO, International Federation of Gynecology Oncology; PCI, peritoneal carcinosis index; EBL, estimated blood loss; ICU, intensive care unit; CC, completeness of cytoreduction.

**Table 2 cancers-13-00523-t002:** Surgical procedures and grade IV complications, N (%) (total 158 patients).

Procedures and Complications	Results
Hysteroadnexiectomies	93 (58.8)
Greater omentectomy	148 (93.7)
Gastric resection	4 (2.5)
Epiploic retrocavity excision	137 (86.7)
Large bowel resection	88 (55.7)
Small bowel resection	15 (9.5)
Total colectomy	14 (8.9)
Splenectomy	38 (24.1)
Right diaphragm S or FTR	118 (74.7)
Left diaphragm S or FTR	69 (43.7)
Glissonian capsule excision	44 (27.8)
Mesenteric root excision	40 (23.3)
Liver metastasectomy	7 (4.4)
Hepatoduodenal ligament peritonectomy	110 (69.6)
Pancreatic resection	4 (2.5)
Ureteral resection	7 (4.4)
Other	4 (2.5)
Grade IV complications	24 (15.2%)
Bleeding	9 (5.7%)
Leak	7 (4.4%)
CHT-induced perforation	2 (1.2%)
Abdominal wall eventration	1 (0.6%)
Pancreatitis	1 (0.6%)
Ileus	1 (0.6%)
Pneumonia	1 (0.6%)
Pelvic abscess	1 (0.6%)
Ureteral leakage	1 (0.6%)

S, stripping; FTR, full-thickness resection; CHT, chemotherapy.

**Table 3 cancers-13-00523-t003:** Univariate and multivariate analysis for overall survival (OS) (total 158 patients).

Variables	Univariate Analysis OS		Multivariate Analysis OS	
	HR (CI%95)	*p* Value	HR (CI%95)	*p* Value
PCI	-	-	-	-
≤15 vs. >15	2.45 (1.59–3.76)	<0.0001	-	-
Cytoreduction	-	<0.0001	-	<0.0001
CC1 vs. CC0	2.84 (1.75–4.63)	<0.0001	2.49 (1.50–4.13)	<0.0001
CC2 vs. CC0	12.07 (6.21–23.25)	<0.0001	9.13 (4.52–18.46)	<0.0001
CC1 vs. CC2	0.24 (0.13–0.44)	<0.0001	0.27 (0.14–0.52)	<0.0001
Timing	-	0.02	-	-
Interval vs. upfront/IDS	2.01 (1.24–3.24)	0.004		
Salvage vs. recurrent	1.15 (0.62–2.13)	0.66		
Diaphragmatic procedure	2.06 (1.14–3.72)	0.02	-	-
Pancreatic resection	11.16 (3.92–31.83)	<0.0001	2.86 (0.95–8.62)	0.06
Total colectomy	2.72 (1.52–4.84)	0.001	-	-
Small bowel resection	2.19 (1.12–427)	0.02	-	-
Number of resections	2.22 (1.39–3.55)	0.001	1.55 (0.94–2.54)	0.08
≥11 vs. <11				
Positive lymph nodes	-	-	-	-

PCI, peritoneal carcinosis index; IDS, interval debulking surgery.

**Table 4 cancers-13-00523-t004:** Univariate and multivariate analysis for disease-free survival (DFS) (total 158 patients).

Variables	Univariate Analysis DFS		Multivariate Analysis DFS	
	HR (CI%95)	*p* Value	HR (CI%95)	*p* Value
PCI	2.04 (1.42–3.04)	<0.0001	-	-
≤15 vs. >15				
Cytoreduction	-	<0.0001	-	<0.0001
CC1 vs. CC0	2.79 (1.83–4.25)	<0.0001	2.63 (1.71–4.02)	<0.0001
CC2 vs. CC0	4.42 (2.51–7.76)	<0.0001	4.22 (2.39–7.42)	<0.0001
CC1 vs. CC2	0.63 (0.36–1.10)	0.1	0.62 (0.36–1.08)	0.09
Timing	-	-	-	-
Interval vs. upfront/IDS				
salvage vs. recurrent				
Diaphragmatic procedure	1.80 (1.12–2.91)	0.02	1.50 (0.93–2.44)	0.01
Pancreatic resection	4.62 (1.68–12.71)	0.003	-	-
Total colectomy	2.56 (1.44–4.53)	0.001	-	-
Small bowel resection	-	-	-	-
Number of resections	1.68 (1.13–2.51)	0.01	-	-
≥11 vs. <11				
Positive lymph nodes	1.67 (1.01–2.76)	0.05	-	-

**Table 5 cancers-13-00523-t005:** Kaplan–Meier for DFS and OS (total 158 patients).

Outcome	Features	% 5 Years	Median (CI 95%)	*p* Values
DFS	Overall	24.3	20 (15–25)	-
	Diaphragmatic Procedures			0.01
	No	37.1	33 (18–47)	
	Yes	20.4	17 (13–20)	
	Cytoreduction			<0.0001
	CC0	38	39 (22–56)	
	CC1	11.1	13 (8–18)	
	CC2	0	9 (6–13)	
OS	Overall	42.1	52 (43–62)	-
	Number of resections			0.001
	<11	51.6	61 (43–80)	
	>11	36.5	39 (30–47)	
	Cytoreduction			<0.0001
	CC0	59	74 (54–94)	
	CC1	31	38 (31–45)	
	CC2	0	15 (7–23)	
	Pancreatic Resections			<0.0001
	No	43.3	53 (46–59)	
	Yes	0	6 (2–10)	

## Data Availability

No new data were created or analyzed in this study. Data sharing is not applicable in this article.

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
