# Peer review of "Cytoreductive Surgery with Hyperthermic Intraperitoneal Chemotherapy for Peritoneal Carcinomatosis from Epithelial Ovarian Cancer: A 20-Year Single-Center Experience"

_cancers, 2021, doi:10.3390/cancers13030523_

Round 1

Reviewer 1 Report

The title states that this manuscript deals with the lessons learned from 20 years’ experience of CRS and HIPEC for epithelial ovarian cancer. However, it shows the outcome of the patients treated in the last 20 years. In total 158 patients are reported over this 20-year period. This means that the centre deals with around 8 patients yearly, which accounts for a low volume centre.  The HIPEC is performed as primary treatment, interval treatment and treatment for recurrence and for a number of biological entities.

The build-up of the article is some what confusing because the method section is placed behind the discussion.

The reader also isn’t informed about the details of the statistical procedures used. Specially is the reader not informed about the starting point of the survival curves.

The presentation isn’t always clear. Table 3 is a combination of more independent calculations, and should therefore be splitted into separated tables. The same is applicable for figure 1 and 2. Figure 3 is confusing, both on presentation and on data.  

In the discussion, we are missing the answer to the primary question of the study: “what have we learned from our 20 years’ experience”.

Above the above mentioned points, it would be nice to see what the authors have changed in the last 20 years and did that result in improvement for the patients.

Author Response

Q. The title states that this manuscript deals with the lessons learned from 20 years’ experience of CRS and HIPEC for epithelial ovarian cancer. However, it shows the outcome of the patients treated in the last 20 years. In total 158 patients are reported over this 20-year period. This means that the centre deals with around 8 patients yearly, which accounts for a low volume centre.

R. The 158 patients decribed represent only those with EOC among a total of more than 400 procedures performed in the same period (approximately 20 each year).

Q. The build-up of the article is some what confusing because the method section is placed behind the discussion.

R. We understand the feeling, but it is the specific editorial format of the journal.

Q. The reader also isn’t informed about the details of the statistical procedures used. Specially is the reader not informed about the starting point of the survival curves.

R. The statistical methods are described in the statistical analysis section inserted after the methods. The missing information has been appropriately added in the methods (P8, I261).

Q. The presentation isn’t always clear. Table 3 is a combination of more independent calculations, and should therefore be splitted into separated tables. The same is applicable for figure 1 and 2. Figure 3 is confusing, both on presentation and on data.  

R. Table 3 has been splitted, as well as Figure 1 and 2 and the description of results has been changed accordingly. Data of Figure 3 have been also rewritten in order to make them clearer (P4, I76).

Q. In the discussion, we are missing the answer to the primary question of the study: “what have we learned from our 20 years’ experience”. Above the above mentioned points, it would be nice to see what the authors have changed in the last 20 years and did that result in improvement for the patients.

R. These are truly appropriate suggestions. Some conclusions of this paper actually resulted from the growing experience during the years, as it occrurred in most similar series, but no statistical data are inserted in order to show it. The beginning of the discussion was then simplified (P4, I82) and the title of the paper was also changed to avoid any potential misunderstanding.

Reviewer 2 Report

Dear Author

I read with interest the manuscript entitled “The Role Of Cytoreductive Surgery With Hyperthermic Intra-2peritoneal Chemotherapy For Peritoneal Carcinomatosis From Epithelial Ovarian Cancer: Lessons Learned From A 20-Year Experience”.

This is the report of a monocentric experience including 158 CRS-HIPEC over 20 years for ovarian peritoneal carcinomatosis. The timing of surgery is heterogeneous.

The subject is very interesting and the experience is large, but it is not sure that this manuscript brings new insights to determine the most appropriate therapeutic strategy for these patients for different reasons detailed thereafter.

Yours sincerely,

First, the retrospective nature of the study with a large period of inclusion impairs the scope of the results.

Secondly, the therapeutic strategy used remain unclear for five important points:

- the policy of adjuvant systemic chemotherapy: it is mentioned that systematic chemotherapy was not systematic after CC-0 resection or when lymph nodes were negative. Is that mean that patients treated with upfront CRS-HIPEC did not always get systemic chemo?

- in the methods, it is mentioned that all patients had retroperitoneal lymph node dissection while it is only half of them in Table 1 and the discussion said that it was done for staging purpose. The discussion part related to lymph node dissection should be clarified avoiding comparing OS on one side and DFS on the other side.

- In the methods part it is mentioned that all patients had preoperative staging laparoscopy while the discussion explained that the introduction of staging laparoscopy allowed a better patient selection. The rate of patients who had a staging laparoscopy before CRS-HIPEC should be mentioned?

- the cohort was made of 85% of FIGO stage IV; It is unusual to have such high rates of stage IV. That outcome should be discussed ++

- the proportion of platinum-resistant patients should be specified

Third, there is a methodological issue: the discussion starts with 2 sentences linked to two results which are not presented in the results chapter.

That leads to a suggestion to improve the manuscript: as the discussion mentioned the learning curve that would be interesting to check if there is a difference in oncological outcomes between the first years (for example the 3 to 5 first years) and the following.

Minor comments

- Abbreviations should be defined when the first time they are used (P3 l65, IDS and PCI)

- P3 l77, remove “the mortality”

Author Response

Q. The subject is very interesting and the experience is large, but it is not sure that this manuscript brings new insights to determine the most appropriate therapeutic strategy for these patients for different reasons detailed thereafter.

R. There is increasing evidence in the literature about the efficacy of CRS with HIPEC in the treatment of advanced EOC, but the precise role is still debated. The purporse of our study was to give a contribution and provide reasons for reflection in order to establish the most appropriate strategy for these patients, as correctly stated by the rewiever. In view of this target, our results clearly showed significant better outcomes for the CC0 resection and Upfront subgroups, at the cost of acceptable postoperative mortaliy and morbidity rates.

Q.First, the retrospective nature of the study with a large period of inclusion impairs the scope of the results.

R. The retrospective nature of the study is clearly a weakness, as declared in the limitations. However, only 1 prospective study on this topic has been reported to date in the literature (Ref. 22).

Q. Secondly, the therapeutic strategy used remain unclear for five important points.

R. The indicated drawbacks have been appropriately clarified has requested.

Q.- the policy of adjuvant systemic chemotherapy: it is mentioned that systematic chemotherapy was not systematic after CC-0 resection or when lymph nodes were negative. Is that mean that patients treated with upfront CRS-HIPEC did not always get systemic chemo?

R. A new sentence has been inserted in order to better explain the concept (P7, I247).

Q.- in the methods, it is mentioned that all patients had retroperitoneal lymph node dissection while it is only half of them in Table 1 and the discussion said that it was done for staging purpose. The discussion part related to lymph node dissection should be clarified avoiding comparing OS on one side and DFS on the other side.

R. A new sentence has been inserted in order to better explain the data (P7, I227).

Q. - In the methods part it is mentioned that all patients had preoperative staging laparoscopy while the discussion explained that the introduction of staging laparoscopy allowed a better patient selection. The rate of patients who had a staging laparoscopy before CRS-HIPEC should be mentioned?

R.The beginning of the discussion has been changed in order to avoid confusion (P4, I86).

Q.- the cohort was made of 85% of FIGO stage IV; It is unusual to have such high rates of stage IV. That outcome should be discussed ++

R. It was a typographical error and we thank the rewiever for having noticed it (Table 1).

Q.- the proportion of platinum-resistant patients should be specified.

R. The missing data has been inserted (P7, I249)

Q. Third, there is a methodological issue: the discussion starts with 2 sentences linked to two results which are not presented in the results chapter. That leads to a suggestion to improve the manuscript: as the discussion mentioned the learning curve that would be interesting to check if there is a difference in oncological outcomes between the first years (for example the 3 to 5 first years) and the following.

R. These are truly appropriate suggestions. Some conclusions of this paper actually resulted from the growing experience during the years, as it occrurred in most similar series, but no statistical data are inserted in order to show it. The beginning of the discussion was then simplified (P4, I82) and the title of the paper was also changed to avoid any potential misunderstanding.

Q.- Abbreviations should be defined when the first time they are used (IDS and PCI)

R. The complete definition of abbreviation has been provided (P3, I64-l65).

Q-, remove “the mortality”

R. The duplicate term has been correctly removed (P3, I77).

Round 2

Reviewer 1 Report

it looks fine now

Author Response

Thank you for approving the changes

Reviewer 2 Report

Dear Authors,

Thank you for your real efforts to respond to our comments.

Your article presents an analysis of factors influencing oncological results but mixing several clinical situations (upfront, interval, recurrence), which precludes strong interpretation and recommendation for clinical application.

For the different survival curves presented, as the population at risk are small, the number of patients at risk as a function of time should be indicated below the curves.

In the conclusion of the abstract, you mentioned that the addition of HIPEC may improve outcomes, however, your work did not bring data comparing population with and without HIPEC. The sentence should probably be rephrased.

The part of the discussion centered on lymph node dissection should be reviewed.

Yours sincerely,

Author Response

Q1. Your article presents an analysis of factors influencing oncological results but mixing several clinical situations (upfront, interval, recurrence), which precludes strong interpretation and recommendation for clinical application.

Q3. In the conclusion of the abstract, you mentioned that the addition of HIPEC may improve outcomes, however, your work did not bring data comparing population with and without HIPEC. The sentence should probably be rephrased.

  1. In our opinion, the answer to these questions should be coupled. We are aware of the limitations, but the major point of interest is exactly to find out the best moment for HIPEC addition in the different timing of treatment for EOC. Large prospective randomized trials would be necessary to definitely answer the question and even if they are difficult to achieve in each arm, ongoing studies (Ref. 25) will help to clarify this issue, as stated in the discussion. Meanwhile, remarkable survival outcomes have been obtained in several retrospective studies, including ours. The sentence in the conclusion of the abstract has been somewhat modified accordingly.

Q2. For the different survival curves presented, as the population at risk are small, the number of patients at risk as a function of time should be indicated below the curves.

  1. The number of patients at risk has been inserted below the different curves.

Q4. The part of the discussion centered on lymph node dissection should be reviewed.

  1. The part of the text focusing on this topic has been shortened and rewritten, since it was actually confusing.

Round 3

Reviewer 2 Report

Thank you for your effort in addressing the comments.

Yours sincerely